# Entomopathogenic Fungi Effectively Control *Phorodon cannabis* Aphid Population in *Cannabis sativa* Plants

**DOI:** 10.3390/plants14060931

**Published:** 2025-03-16

**Authors:** Daniel Lopez Restrepo, Igor Kovalchuk

**Affiliations:** Department of Biological Science, University of Lethbridge, Lethbridge, AB T1K 3M4, Canada; daniel.lopezrestrepo@uleth.ca

**Keywords:** *Cannabis sativa*, cannabis aphids, *Phorodon cannabis*, entomopathogenic fungi, *Beauveria bassiana*, *Metarhizium anisopliae*, aphid control with fungi

## Abstract

The rapid expansion of the cannabis industry in Canada post-legalization has heightened the prevalence of pests, particularly the cannabis aphid *Phorodon cannabis* (*P. cannabis*), which poses significant threats to crop health. This study investigates the immediate effects of *P. cannabis* on *Cannabis sativa* (*C. sativa*) plants and explores biological control strategies utilizing entomopathogenic fungi. Fungal isolates of *Beauveria bassiana* and *Metarhizium anisopliae* were isolated from infected aphids, cultured, and characterized. Infection tests on aphids revealed that both fungi achieved 100% aphid mortality at high conidial concentrations (1 × 10^7^ conidia/mL) by the 10th DAT, with *Beauveria bassiana* demonstrating better efficacy. In greenhouse trials on three cannabis varieties, *B. bassiana* effectively controlled aphid populations, keeping levels low and stable in infested plants treated with *B. bassiana* at the concentration of 1 × 10^7^ conidia mL^−1^ (I-B) and infested plants treated with insecticide (I-I). Both I-B and I-I treatments maintained aphid populations near zero for nine weeks. In contrast, control plants showed significant aphid growth, with the Perseid variety being the most susceptible, followed by Congo Durban, while GCC exhibited the lowest susceptibility. Cannabinoid and terpene analyses revealed that treatment with insecticide substantially decreased the amount of produced cannabinoids and terpenes. In contrast, *Beauveria bassiana*-treated plants exhibited higher concentrations of key metabolites, including delta-9-tetrahydrocannabinolic acid and cannabidiolic acid, and total terpenes, compared to chemically treated plants, and in two out of three cultivars, these concentrations were higher than in control, untreated plants. The findings highlight *Beauveria bassiana* as an eco-friendly alternative for pest management that not only controls aphids effectively but also supports the biochemical quality of cannabis plants.

## 1. Introduction

Cannabis cultivation in Canada has evolved significantly, shifting from primarily industrial uses to becoming a key agricultural sector following the legalization of recreational cannabis through the Cannabis Act in 2018 [1,2]. Most cannabis cultivation in Canada occurs indoors or in greenhouses, utilizing advanced techniques such as hydroponics, aeroponics, and controlled-environment agriculture to optimize quality and sustainability [3].

A key challenge that cannabis growers face is the emergence of new pests, particularly *Phorodon cannabis*, or the cannabis aphid, which weakens plants by feeding on their phloem. This feeding behavior disrupts the flow of essential nutrients and sugars, ultimately reducing plant growth and vigor. Aphid infestation can lead to stunted growth, yellowing of leaves, and overall poor plant health, which compromises the quality and yield of cannabis crops [4,5]. The honeydew excreted by these aphids also promotes the growth of pathogenic fungi, further exacerbating plant stress and creating an environment that attracts other insects, such as ants [6]. Additionally, the accumulation of aphid exoskeletons on the plant surface can obstruct photosynthesis, further hindering plant development and reducing overall crop productivity. With limited knowledge about the long-term effects of cannabis aphid infestations and limited pest control options, sustainable solutions are urgently needed. Current methods rely heavily on chemical insecticides, which pose risks such as environmental damage, chemical residues, and pest resistance [7]. Integrated pest management (IPM) strategies that incorporate biological control methods, such as entomopathogenic fungi (EPF), offer a promising alternative to reduce *P. cannabis* populations sustainably [8].

EPF are naturally occurring fungi that act as insect pathogens, making them a viable alternative to chemical pesticides [9]. These fungi infect pests by penetrating their cuticle using hydrolytic enzymes and specialized structures, ultimately killing the host [10]. Commercial EPF strains such as *Beauveria bassiana*, *Metarhizium anisopliae*, and *Akanthomyces lecanii* are available in Canada; however, their application for controlling *P. cannabis* remains underexplored [11]. EPF offer several advantages in pest management. First, their high specificity minimizes harm to beneficial insects and ecosystems [8,12]. Second, their low toxicity to mammals and biodegradability reduce environmental and health risks, unlike chemical pesticides that often persist in the environment [13,14]. EPF have demonstrated efficacy in reducing aphid populations in various crops, improving plant health and yield while lowering reliance on chemical pesticides [15,16].

Research has demonstrated the efficacy of entomopathogenic fungi such as *B. bassiana* and *Akanthomyces lecanii* against aphids. For instance, *B. bassiana* achieved over 80% mortality in aphid populations within six days in controlled conditions, with virulence influenced by environmental factors like temperature, humidity, and UV light [17]. Another study showed that *B. bassiana* inoculation in cucumber plants increased secondary metabolites, reducing the fitness and population growth of cotton aphids (*Aphis gossypii*) by altering herbivore–plant interactions and reducing aphids’ detoxifying enzyme activity [18]. Further, *B. bassiana* strains achieved 74% aphid mortality within seven days in field and greenhouse trials, confirming its potential as a biocontrol agent for aphids in crops such as cowpeas [19]. Similarly, *M. anisopliae* demonstrated 97% mortality against *A. gossypii* at specific conidia concentrations in *Capsicum annum* and *Solanum melongena* crops, showcasing its effectiveness across different host plants and aphid life stages [20].

This research presents a novel approach to managing cannabis aphids through the application of EPF, specifically *Beauveria bassiana*. Given the limited existing studies on the biological control of *P. cannabis*, we hypothesized that entomopathogenic microorganisms are efficient biocontrol agents for *Phorodon cannabis* in *Cannabis sativa* seedlings under greenhouse conditions. We isolated two strains of entomopathogenic fungi and demonstrated that they control aphid populations as efficiently as commercial insecticides. We also show that infestation with aphids affects the cannabinoids and terpene profiles and that the application of insecticide significantly reduces the concentration of these compounds, while treatment with *Beauveria bassiana* prevents this negative change. This work significantly expands the knowledge base by demonstrating the efficacy of *B. bassiana* in reducing aphid populations while maintaining plant health. Furthermore, it uniquely explores the impact of aphid infestation and its biological control on the production of cannabinoids and terpenes in *Cannabis sativa*.

## 2. Results


**Isolation of entomopathogenic fungi**


After obtaining samples of dead or infected insects from the Hepler Hall greenhouse, fungal isolates were obtained following a seven-day incubation period of serial dilutions of enriched samples, which were then plated onto Potato Dextrose Agar (PDA). A total of 20 fungal isolates were recovered after purification and subculturing. Subsequently, lactophenol blue staining was applied to enhance the visibility and contrast of fungal structures under the microscope, making key morphological features easier to observe. The stain binds to chitin in the fungal cell walls, highlighting hyphae, spores, and conidiophores with a blue tint against a clear background.

Among the isolates obtained, colonies consistent with *Beauveria* (F2), *Metarhizium* (F12), *Trichoderma* (F17, F18), *Penicillium* (F1), and *Aspergillus* (F10, F16) were identified (Appendix A).

Molecular analysis confirmed the morphological identification, revealing that the two EPF isolates (F2 and F12) obtained from insect cadavers were a *Beauveria bassiana* isolate and a *Metarhizium anisopliae* isolate, respectively. This identification was achieved by sequencing the internal transcribed spacer (ITS) region of the rDNA using the ITS1 and ITS4 primers. Phylogenetic analysis, conducted using the Neighbor-Joining method in MEGA 11 (Figure 1), produced a bootstrap consensus tree. The sequences of this isolate showed 99% homology with another *B. bassiana* and *Metarhizium anisopliae* isolates in the GenBank database.


**Establishment of cannabis aphid**
**colonies on cannabis plants**


Aphid colonies were successfully established on cannabis plants of the Congo Durban variety, and new plants had to be infested approximately every two months to maintain the colonies. The aphid infestation spread rapidly on the plant, and a new plant had to be infested about every month. Production of large amounts of honeydew was evident, and obvious damage to the plants (e.g., wilting, yellowing) was observed (Figure 2A,B).

Using a stereo microscope at 20× magnification, we determined the morphological characteristics of the aphids to discriminate *P. cannabis* from other aphid species (Figure 2C,D). It was possible to identify the aphid as *P. cannabis* and distinguish it from the most similar species, *Phorodon humuli* (hop aphid), by key morphological features. *P. cannabis* has flabellate (spatulate) body setae, slightly swollen siphunculi, and shorter antennal segment III relative to the siphunculi. In contrast, *P. humuli* lacks flabellate setae and has straight siphunculi and a longer antennal segment III [4]. Also, *P. cannabis* has a slightly lighter color than *P. humuli*, especially evident in adults. These differences provide reliable markers for distinguishing between the two species on cannabis plants.


**Entomopathogenic bioassay**


The pathogenicity test of the isolates against *P. cannabis* was carried out using the leaf-dip method with some modifications. Figure 3A shows the mean mortality rate of *P. cannabis* on different days after treatment (DAT) with the entomopathogenic fungi *B. bassiana* and *M. anisopliae* across two conidial concentrations (1 × 10^5^ and 1 × 10^7^ conidia mL^−1^), plus a control group.

At the highest concentration (1 × 10^7^), both fungi achieved a 100% mortality by the 10th DAT; however, there were no statistically significant differences between the two microorganisms (*p* > 0.05). Lower concentrations and earlier DATs generally resulted in lower mortality, highlighting dose- and time-dependent effects on aphid mortality. However, with the concentration of 1 × 10^7^ conidia mL^−1^ *B. bassiana* reached a higher mean mortality (*p* < 0.05) by the 7th DAT, showing a higher infection rate compared to *M. anisopliae* at the same concentration and the same DAT. The average aphid mortality of the control treatment only increased starting on the 7th DAT and only reached 22% on the 10th DAT of treatment. It is because of these results that it was decided to use the *B. bassiana* inoculum at a concentration of 1 × 10^7^ conidia mL^−1^ for the greenhouse bioassay.

The dead aphids were placed on a new PDA plate, incubated, and then analyzed under the microscope to observe the growth of the fungal mycelium as a cause of death. On the 3rd DAT, the first signs of infection were evident: fungal spores adhered and germinated in the aphid cuticle, with small germ tubes that began to penetrate the insect’s exoskeleton.

Later, on the 5th DAT, it was seen that the infection was progressing; filiform hyphae were spreading throughout the aphid, consuming nutrients and altering its tissues. After this, on the 7th and 10th DAT, hyphae grew outward from the aphid’s body, breaking the cuticle and covering the insect with a fuzzy white layer of fungal mycelium (Figure 3B–E). This external growth produces new spores on the surface of the aphid, ready to infect other insects, demonstrating *B. bassiana*’s and *M. anisopliae*’s effectiveness as biological pest control agents.


**Effects of aphid infestation on cannabis plant growth parameters**


To analyze the effect of aphid infestation on cannabis plants and to evaluate the efficiency of entomopathogenic fungi in controlling it, we set up the following treatments. Control— water-treated only (Ct), infested with aphids, treated with water (I-W), infested with aphids, treated with commercial insecticide Doktor Doom (I-I), infested with aphids, treated with *B. bassiana* (I-B), and non-infested, treated with *B. bassiana* (N-B). For all treatments, we used three cultivars: a THC-dominant cultivar, Congo Durban; a CBD-dominant cultivar, Perseid; and a balanced THC/CBD cultivar, GCC.

Approximately 9 weeks after the flowering process began, the three cannabis varieties were dried, weighed, and measured to determine the growth parameters of the treatments. Figure 4 shows the growth parameters (dry biomass and height) of all three varieties under different treatments. The dry biomass (Figure 4A) of Ct (uninfested and treated with distilled water) plants was the largest in Perseid and GCC, with GCC plants being the heaviest, ~9 g (Figure 4A). I-W and I-I plants had the lowest biomass in all three cultivars, with the exception of I-W in Congo Durban, although a significant difference was only observed in Perseid and GCC cultivars (*p* < 0.05). There was no significant difference between I-B and I-I plants, although, in Congo Durban, the I-I group was notably lighter, suggesting similar or better efficacy of *B. bassiana* over insecticide in mitigating biomass loss by aphid infestation.

Ct plants were the tallest in all three cultivars. Aphid infestation reduced plant height across treatments, with I-W plants being the shortest; the difference was significant in Congo Durban and Perseid but not GCC (Figure 4B,D,F). *B. bassiana* and the chemical insecticide helped maintain greater height compared to I-W, but Bb treatments were similar to I-I (Figure 4B,D,F).

In summary, *P. cannabis* infection significantly reduced both height and dry biomass in cannabis plants, particularly when untreated. While Ct, I-I, and I-B plants showed similar growth outcomes in height and biomass, I-W plants exhibited the lowest values for both metrics. These findings highlight that *P. cannabis* infection can substantially impair plant growth and productivity if left unmanaged, emphasizing the importance of appropriate pest control measures to maintain healthy plant development.


**Effects of *Beauveria bassiana* on aphid population**


Since *B. bassiana* (Bb) appeared to have a more substantial effect over *Metarhizium anisopliae*, we next tested the effect of *B. bassiana* (Bb) on aphid population in three infested cultivars. The aphid populations remained low and stable in I-B and I-I plants in all three tested cultivars (Figure 5). The Bb and chemical insecticide treatments effectively controlled aphid population growth, maintaining it close to zero throughout the 9-week period after infestation, with minimal variation.

Across all varieties, I-W plants experienced significant aphid population growth over the 9-week experimental period. The highest unchecked increase in aphid population was observed in Perseid, indicating its highest susceptibility to infestation; Congo Durban had lower susceptibility, and GCC had the lowest (Figure 5C).


**Effects of aphid infestation and treatment with *B. bassiana* on cannabinoid concentration**


We next measured cannabinoid content in all groups. In the THC-dominant cannabis variety Congo Durban, the total cannabinoid content was the highest in N-B plants (29.52%), closely followed by the I-B group (28.72%) (Figure 6A). I-I had the lowest total cannabinoid concentration (24.39%), suggesting that chemical insecticide treatment might suppress overall cannabinoid production, while Bb treatment supports higher total cannabinoid levels, even with infestation. The effect of infestation and treatment was also seen on individual cannabinoids, where THCa and Δ9-THC were highest in I-B and N-B as well as Ct plants and lowest in I-I plants. A similar trend was observed for minor cannabinoids.

A similar response was observed in the CBD-dominant Perseid variety; the total cannabinoid concentration was also the highest in the I-B (15.71%) and N-B (15.45%) plants, further suggesting that *B. bassiana* significantly enhances the total cannabinoid content, particularly in infested plants, compared to the other treatments, for example, I-I plants (13.32%) (Figure 6C). Again, a similar trend was observed for individual cannabinoids.

In the balanced cultivar GCC, the response to treatments was a bit different than in Congo Durban or Perseid. The total cannabinoid concentration was, surprisingly, the highest in I-W plants (28.93%) (Figure 6E). The I-I group still had the lowest total cannabinoid concentration (18.07%), again indicating a suppressive effect of insecticide on cannabinoid biosynthesis. The total cannabinoid content in the Ct, I-B, and N-B groups was comparable, suggesting that Bb treatment did not interfere with cannabinoid production. Again, individual cannabinoids followed the trend.

Since we observed the effect of aphid infestation on the total dry weight of plants, we decided to calculate the total amount of produced cannabinoids in individual plants by multiplying plant weight by cannabinoid concentration. Congo Durban I-B plants had the highest weight of total cannabinoids (2.33 g), followed by Ct (2.13 g) and N-B plants (1.99 g) (Figure 6B). I-I plants had the lowest total cannabinoid production (1.69 g). This was also reflected in THCa and Δ9-THC and minor cannabinoids that were also highest in the I-B, N-B plants, and Ct plants, while lowest in the I-I plants.

In the Perseid cultivar, total cannabinoid weight was the highest in I-B plants (2.33 g), followed by N-B and Ct plants; I-W plants produced the lowest amount of cannabinoids by weight (Figure 6D). A similar picture was observed for CBDA and THCA.

In GCC, the total cannabinoid production was highest in Ct plants (2.39 g), followed by I-B (2.20 g), I-W (1.95 g), and N-B (1.89 g) plants (Figure 6E). The I-I group produced the lowest amount of cannabinoids (1.30 g).

This cannabinoid profile analysis suggests that infestation with aphids decreases the cannabinoid content. Also, it shows that treatment with insecticide decreases total and individual cannabinoid content in all varieties. Finally, in Congo Durban and Perseid varieties that are more sensitive to aphids, *Bb* treatment prevents the drop-in cannabinoid content and even increases it compared to Ct. In contrast, in the GCC variety that is less sensitive to cannabinoids, *Bb* treatment also prevents the drop-in cannabinoids compared to insecticide but does not increase it in comparison to the water treatment.


**Effects of aphid infestation and treatment with *B. bassiana* on terpene concentration**


The terpene profile analysis of Congo Durban cannabis plants reveals notable differences in total terpene concentrations (Figure 7A). N-B plants exhibit the highest terpene concentration at 21.19 mg/g, suggesting that Bb treatment might positively impact terpene production. I-W plants displayed a terpene concentration of 19.86 mg/g, indicating that while infestation affects the plant, infestation does not drastically reduce terpene levels. I-B plants showed a terpene concentration of 16.96 mg/g, while I-I had the lowest terpene concentration, at 13.49 mg/g. This suggests that chemical treatments may inhibit terpene synthesis or negatively impact overall terpene profiles, while Bb treatment results in a lower decrease in terpene synthesis. Finally, Ct plants had a terpene concentration of 16.22 mg/g, which is lower than that of N-B and I-W, comparable to I-B, and much higher than I-I (Figure 7A).

A similar picture was observed in two other cultivars, Perseid (Figure 7C) and GCC (Figure 7E), where the N-B group had a higher concentration of terpenes, followed by N-W and I-B plants; Ct plants had terpene levels comparable to I-B. The I-I was the lowest by far in both cultivars.

Since we observed the effect of infestation with aphids and different treatments on plant biomass, we decided to calculate the total and individual weight of produced terpenes by multiplying their concentrations by the plant dry weight (Figure 7B,D,F). In Congo Durban, the I-B group had the largest amount of terpenes produced, and I-I had the lowest, with other groups being in between and similar to each other. In fact, in Congo Durban, treatment of plants with Bb allowed them to produce more terpenes than control or I-I groups (Figure 7B). In Perseid, the largest amount of terpenes was also produced in N-B and I-B groups, followed by Ct; the amounts of terpenes in I-I and I-W were much lower (Figure 7D). In GCC, the N-B group had the largest amount of terpenes, followed by I-W, Ct, and then I-B; again, the I-I group had the lowest amount (Figure 7F).

As far as individual terpenes are concerned, there was a change in nerolidol, fenchone, isoborneol, and alpha-bisabolol in response to treatments. Specifically, nerolidol in Congo Durban was undetectable in Ct plants but increased to high levels in I-W and I-B-treated plants; similarly, in Perseid, it increased in I-B and N-B plants (Appendix A). Isoborneol also increased substantially in I-W Perseid plants (Appendix A). Fenchone decreased strongly (beyond the detection limit) in I-W plants but was present again after treatment with insecticide and Bb in Congo Durban (Appendix A). Alpha-bisabolol was reduced dramatically in I-W plants and was still low in all treatments in GCC plants (Appendix A). We also noted that in the Perseid cultivar, infested plants treated with water (I-W) and insecticide (I-I) had substantially decreased levels of alpha-pinene, beta-myrcene, beta-pinene, guaiol, alpha-cedrene, and delta-limonene (Appendix A).

These results highlight a potential downside to using chemical insecticides on cannabis plants, as they might suppress the expression of key secondary metabolites like terpenes and cannabinoids. Thus, Bb appears to be a favorable pest management alternative that may enhance or sustain terpene production, even in the presence of aphid infestation, compared to chemical insecticides, which could reduce the aromatic and therapeutic quality of the cannabis product.

## 3. Discussion


**Entomopathogenic bioassay**


The pathogenicity results of *B. bassiana* and *M. anisopliae* against *P. cannabis* are consistent with studies of EPF as biological control agents for aphids [21]. At a concentration of conidia 1 × 10^7^, both isolates demonstrated high effectiveness, with 100% mortality by the 10th DAT. Previously, studies on *Aphis gossypii* [22] and *Aphis craccivora* [15] treated with EPF, including *B. bassiana* and *M. anisopliae*, reported mortality rates of 80–100% at comparable concentrations and timelines, emphasizing dose- and time-dependent effects on pest mortality.

The faster infection progression and higher mortality rate by *B. bassiana* at 7 DAT in our study are supported by prior evidence suggesting that *B. bassiana* often has a slightly faster infection cycle compared to *M. anisopliae*. For instance, research on diamondback moth larvae and aphid species observed a quicker onset of mortality with *B. bassiana*, although *M. anisopliae* displayed comparable effectiveness over longer durations [23]. Another study [24] showed that while *M. anisopliae* performs comparably to *B. bassiana* over time, its efficacy may depend more on specific environmental conditions, such as humidity and temperature. In field evaluations, both fungi successfully reduced pest populations, but *B. bassiana* often achieved faster initial mortality and higher spore viability on the insect host. These findings underscore *B. bassiana*’s potential as a faster-acting biocontrol agent under controlled conditions.

Microscopic observations of fungal development corroborate the mode of action of these fungi. Initial spore adhesion and germination on the aphid cuticle by the third DAT, followed by extensive tissue colonization and hyphal outgrowth, are hallmark features of EPF pathogenicity. These processes mirror findings in other reports, where fungal growth from insect cadavers produces new infectious conidia, underscoring the fungi’s potential for population-level suppression of aphids [25].


**Effects of aphid infestation on cannabis plant growth parameters**


The comparable effectiveness of *B. bassiana* and the chemical insecticide we used in mitigating biomass loss highlights the potential of EPF as a sustainable alternative in IPM. Similar studies on tomatoes and other crops have reported that *B. bassiana* not only controls pest populations but can also enhance plant resilience and growth, sometimes even stimulating beneficial traits such as root and shoot development under certain conditions [26,27]. Root length was significantly greater at 3, 4, and 5 days post-sowing (*p* < 0.05), and plant height was significantly higher at 7, 14, and 21 days post-emergence. Field experiments demonstrated improvements in tomato yield. Fruit quantity under the Bb treatment increased by 22.9–28.0% compared to controls and *Bacillus cereus*. The findings suggest that *B. bassiana* enhances tomato growth and yield characteristics [26]. The fungal colonization of plant tissues, a phenomenon referred to as endophytism, may contribute to reduced pest-induced stress and improve nutrient uptake.

The modest reductions in plant height and biomass in infested cannabis plants treated with *B. bassiana* and chemical insecticides suggest that while these treatments manage aphid populations effectively, they may not fully reverse the stress induced by initial infestations. This pattern mirrors findings in tomato and maize crops [26], where biocontrol agents reduced pest damage, but plant growth metrics were slightly lower compared to uninfested controls.


**Effects of *B. bassiana* on aphid population**


Bb is a widely studied entomopathogenic fungus that infects and kills a range of insect pests, including aphids, through spore adhesion and subsequent colonization of the insect cuticle [10]. Studies have shown that Bb formulations can effectively reduce pest populations in various crops [18], supporting its performance in controlling *P. cannabis* observed in the current study.

The low and stable aphid populations observed in Bb-treated plants of all three cannabis varieties are consistent with findings in other studies, where Bb achieved effective and sustained control of aphid populations under greenhouse conditions [28,29].

Our study also showed varietal differences in *P. cannabis* susceptibility. Untreated plants showed significant population increases, with Perseid experiencing the most rapid growth (exceeding 350 aphids by week 9). This aligns with prior research suggesting that host plant characteristics, such as nutrient composition and secondary metabolite profiles, can influence aphid reproduction and survival [30]; this report showed that plants with higher amounts of essential oil extracts, which are rich in terpenoids, were more effective against infestation by small arthropods like mosquitoes, aphids, and spider mites. The relatively moderate growth in GCC and slower growth in Congo Durban may be attributed to varietal differences in resistance mechanisms, such as metabolite defenses, which have been documented in related crops [30].


**Effects of aphid infestation and treatment with *B. bassiana* on cannabinoid and terpene concentration**


For the THC-dominant Congo Durban variety, Bb treatment significantly increased THCa and Δ9-THC levels, particularly in non-infested plants, where the highest concentrations of THCa (19.68%) and Δ9-THC (7.07%) were observed. Infested plants treated with the chemical insecticide showed the lowest concentrations of these cannabinoids, with THCa at 16.52% and Δ9-THC at 5.66%. These findings align with previous studies that suggest that biocontrol agents can elicit induced systemic resistance in plants, activating metabolic pathways associated with secondary metabolite production [31].

Although there is not much research on the interactions of EPF with cannabis plants to facilitate understanding of these interactions, there are reports on other microorganisms and their direct interactions with host metabolism. Endophytic fungal isolates have been shown to confer disease resistance against virulent pathogens, nematodes, and insects, and this resistance has been correlated with increased concentrations of phenolic metabolites [32].

In the balanced GCC variety, the highest total cannabinoid content (28.93%) was found in water-treated infested plants, suggesting that natural stress responses might enhance secondary metabolite synthesis, as has been documented in studies on stress-induced biosynthesis of cannabinoids and terpenes [33]. However, Bb treatments maintained robust cannabinoid profiles, with total concentrations around 25.52–25.68%, despite aphid infestation. Treatment with insecticide, however, resulted in the lowest cannabinoid content (18.07%), as was observed in the other two varieties, mirroring findings in other crop systems where synthetic insecticides negatively impacted secondary metabolites due to their interference with key enzymatic pathways [34,35].

Additionally, the manufacturing process for cannabis products, such as oils and concentrates, can lead to a concentration of pesticide residues in the final extracts. In some cases, highly toxic pyrolysis products may form exclusively from pesticide residues during the smoking process [36]. Such risks highlight the importance of minimizing pesticide use in cannabis cultivation. By contrast, EPF provide a safer, eco-friendly solution that aligns with the need to maintain cannabinoid and metabolite integrity while protecting consumer health.

The observed increase in terpene concentrations in Bb-treated plants, even under aphid infestation, suggests that biological control agents can be used to enhance plant secondary metabolite production. The appearance of some and the disappearance of other compounds in response to infestation or/and treatment with insecticide could change the potency of cannabis flowers and their effect on humans, especially those who use it for medicinal purposes. These results align with findings from other researchers [37] who showed that insect-feeding damage induced cotton plants to synthesize novel volatile compounds. The source and nature of the factor(s) responsible for both increased production and release of compounds normally produced by the plants, as well as biosynthesis of the compounds not observed in untreated plants, are not known but likely represent the plant defense mechanism [38].

## 4. Materials and Methods


**Isolation of entomopathogenic fungi**


Infected insect samples were collected and stored in hermetically sealed plastic bags. For the isolation of entomopathogenic fungi, each sample was enriched in sterile distilled water supplemented with 0.1% peptone, 0.5% Tween 80, and 0.05 mg/mL chloramphenicol; these were incubated for two hours at room temperature and 100 rpm agitation on a rotatory shaker. Serial dilutions were prepared to cultivate and isolate fungi and then plated on potato dextrose agar (PDA: 200 g/L potato extract, 20 g/L dextrose, and 15 g/L agar in distilled water pH 3.5). After seven days of incubation, the single spore culture method was used to purify fungal cultures. A fungal suspension was prepared, and a sterile needle was dipped into the suspension and streaked onto a solid agar plate. The plates were incubated for several days to allow individual spores to develop into distinct colonies. Selected spores were then recultivated on a new solid plate to obtain purified fungal isolates and then stored for future research. Subsequently, lactophenol blue staining was performed to enhance the visibility and contrast of fungal structures under a microscope, making key morphological details easier to observe. The stain binds to chitin in the fungal cell walls, highlighting hyphae, spores, and conidiophores with a blue tint against a clear background, as well as morphological keys of the colonies. Fungal samples were first observed for colony color, texture, and growth patterns on Potato Dextrose Agar (PDA) plates. Photos of lactophenol blue staining were taken under a microscope at 40× magnification. In addition, photos of the fungal colonies were taken in the petri dishes after seven days of incubation. Colony colors, such as white, green, or cream, and textures, such as cottony, powdery, or granular, were recorded as preliminary identifiers to select isolates consistent with entomopathogenic fungi of the genera *Beauveria*, *Metarhizium*, and *Akanthomyces*.


**Molecular characterization of the entomopathogenic fungi isolates**


The isolates were classified by molecular analysis, for which DNA extraction was performed using the NORGEN Plant/Fungi DNA isolation kit (Cat. 27300). At the end of the extraction process, DNA quantification was performed using the 260 nm light absorption method (Nanodrop). For PCR, the oligos ITS1: 5′TCCGTAGGTGAACCTGCGG 3′ and ITS4: 5′TCCTCCGCTTATTGATATGC 3′ were used, which amplified a fragment around 750 base pairs (bp). The PCR products were sent to Eurofins Genomics for sequencing using the Sanger method. Forward and reverse sequences were obtained, refined, edited, and aligned. The consensus sequences were compared with the available sequences in the GeneBank database using nucleotide BLAST (version 2.16.0+/25 June 2024) to determine the identity of the fungal isolates. Finally, the phylogenetic tree of the isolates was constructed using the Neighbor-Joining method integrated in MEGA 11.


**Preparation of the inoculum solution of entomopathogenic microorganisms**


The isolates were grown on PDA at 25 ± 1 °C for 15 days. Conidia were harvested with sterile distilled water containing 0.05% Tween 80. Mycelia were removed by filtering conidia suspensions through 4 layers of sterile cheesecloth. Conidia were counted under a microscope using a Neubauer hemocytometer to adjust the suspension concentration to 1 × 10^7^ cells/mL for each isolate.


**Cannabis aphids**


*Phorodon cannabis* insects were obtained from infested plants in Hepler Hall at the University of Lethbridge during the spring of 2023. Insects were isolated and cultivated in another greenhouse, and later, some wingless females were transferred to cannabis plants of the variety “Congo diesel” to establish colonies. The plants were fed regularly with FloraGro, which is rich in nitrogen, during the vegetative stage to promote healthy leaf and stem development. As the plants transitioned to flowering, FloraBloom, high in phosphorus and potassium, was introduced to support bud growth and root strength. Throughout both stages, FloraMicro was applied to provide essential trace elements, and defoliation was carried out periodically to control the population and infestation with aphids. Old plants were replaced with new ones every time that was necessary. The insects were observed using a stereo microscope at 20× magnification to determine the morphological characters to discriminate *Phorodon cannabis* from other aphid species [39].


**Entomopathogenic bioassay**


The pathogenicity test of the isolates against *P. cannabis* was carried out using the leaf-dip method with some modifications [40]. For each microorganism, cannabis leaf discs of 50 mm diameter were obtained from healthy plants and immersed for 10 s in 5 mL of fungal suspensions obtained as described above. Then, to reduce the amount of excessive fungal suspension, the leaves were placed on sterile filter paper for 15 min and taken to sterile petri dishes containing 1% agar solution. As a control, leaf discs immersed in 0.05% Tween 80 were used. Subsequently, non-winged adults were collected from infested plants and transferred into the leaf discs. The petri dishes were incubated at room temperature (25 °C) in the dark for 10 days. Each treatment was repeated ten times, and data were taken on the mortality of the aphids on days 3, 5, 7, and 10 of incubation, during which time, the dead aphids were removed, placed on a new PDA plate, incubated, and then analyzed in the microscope to observe if the growth of the mycelium of the fungus was the cause of death.


**Greenhouse bioassay**


Cuttings from mother plants, approximately 6 months old, were rooted for 10 days, after which the plants were transferred to pots with a mixture of peat moss and perlite 70:30 ratio. Plants were grown at 22 °C with an 18 h light 6 h dark cycle for 4 weeks and then transferred to chambers with a 12 h light/12 h dark regime to promote flowering. Five treatments were applied to the cannabis plants: (1) uninfested, treated with distilled water; (2) uninfested, treated with entomopathogenic microorganism; (3) aphid-infested, treated with commercial chemical insecticide according to the manufacturer’s specifications (Doktor Doom Premium Indoor/Outdoor 3-in-1 Crop and Plant Rescue Concentrate, Doctor Doom, Edmonton, Canada; active ingredient is permethrin); (4) aphid-infested treated with distilled water; and (5) aphid-infested, treated with entomopathogenic microorganism. Wingless adult aphids were inoculated on each plant according to the treatment. One week after the inoculation of the aphids and using an atomizer, the plants were foliar sprayed with the fungal solution (1 × 10^7^ cells/mL) as well as the chemical insecticide or the control with distilled water only according to each treatment. Spraying was repeated once a week until the flowering process was completed [15]. By the end of the flowering period, the flowers were harvested, dried, weighed, measured, and later sent for subsequent analysis by HPLC at Canvas Labs (Vancouver, BC, Canada) to determine the concentrations of important cannabinoids and terpenes.


**Data collection**


To evaluate how effective entomopathogenic microorganisms are in controlling *P. cannabis* within greenhouse conditions, the aphid population was determined for each treatment described above.

Samples of three leaves were taken from each plant, one from the top, the middle, and the base of each plant, before and after each treatment. Data were taken on the number of live aphids in each sample, and the population was calculated each week for each treatment [41].

To assess the efficacy of pest control measures by comparing growth parameters between plants subjected to treatment and those left untreated, the weight and height of the plants (base of the pylon to the highest point of the crown of the leaves) were measured. Dry biomass was determined after flowering; for this, each plant was taken to an oven for 72 h at 60 °C, after which each plant was weighed on a top-loading laboratory balance [42].


**Statistical analysis**


The response variables were initially described through descriptive statistics and, later on, analyzed using parametric tests. In all cases, the variables were expressed as the mean ± standard deviation. To determine the effects of the different treatments on the response variables, a one-way ANOVA was implemented, followed by a Tukey test. Before any statistical analysis, Brown and Forsythe’s tests were performed to determine the homogeneity of variance, and the Kolmogorov–Smirnov test was performed to test the normality of the obtained data, with the aim of verifying compliance with these two assumptions without which the analysis did not proceed. In case of non-compliance with the above assumptions, the data in percentages were subjected to arc-sine transformations to be normalized, or ultimately, they were analyzed using the non-parametric Kruskal–Wallis test. In all cases, *p* < 0.05 was used as a statistical criterion to reveal significant differences between treatments, considering a 95% test confidence interval. All the data were analyzed using the IBS SPSS Statistics version 25 statistical program.

## 5. Conclusions

Overall, Bb demonstrated a consistent ability to sustain or enhance secondary metabolite production in cannabis under both pest-free and aphid-infested conditions. The results suggest that Bb leads to increased synthesis of cannabinoids and terpenes, thereby likely improving the plant’s aromatic and therapeutic properties. Conversely, the chemical control used in this research suppressed secondary metabolite production across all varieties, potentially through their negative impacts on plant metabolism and stress-response pathways, as has been reported in agricultural studies [14,43]. These findings highlight the potential of Bb as a sustainable pest management strategy in cannabis cultivation, offering dual benefits of pest control and quality enhancement. Future research should focus on elucidating the molecular mechanisms underlying these effects and optimizing Bb formulations and application protocols for large-scale cannabis production. Refining environmental parameters such as temperature, humidity, and light conditions will be essential to maximize fungal spore germination and infection efficacy. Developing advanced formulations, such as oil-based or encapsulated spores, could improve the life and application effectiveness of EPF. Further investigations should also explore their integration with other IPM strategies, including the use of predatory insects, insecticidal soaps, and resistant cannabis cultivars, to create synergistic and holistic approaches to pest control.

## Figures and Tables

**Figure 1 plants-14-00931-f001:**
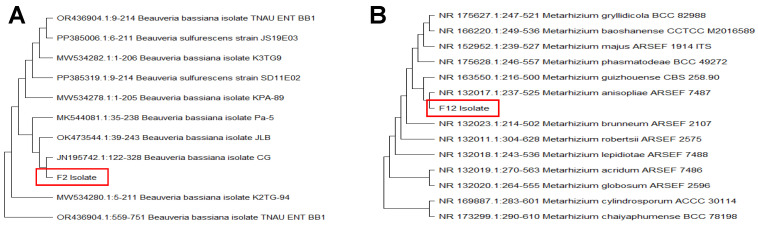
Phylogenetic tree for fungal isolate F2 *Beauveria bassiana* (**A**) and F12 (**B**) *Metarhizium anisopliae* based on the ITS sequences. Tree constructed using the Neighbor-Joining method integrated in MEGA 11. F2 and F12 isolates are in the red box, for emphasis.

**Figure 2 plants-14-00931-f002:**
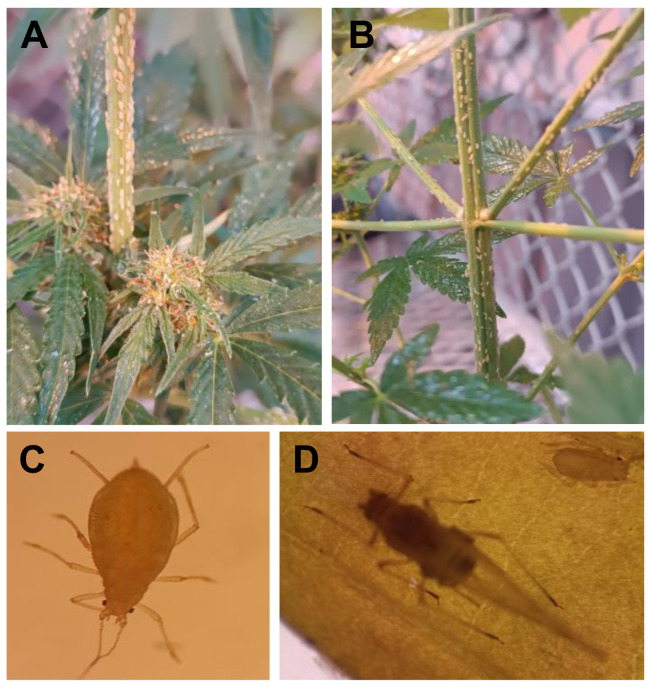
Cannabis plants of the Congo Durban variety (**A**,**B**) infested with *P. cannabis*. (**C**) Adult aphid. (**D**) Winged adult, under a stereo microscope at 20× magnification.

**Figure 3 plants-14-00931-f003:**
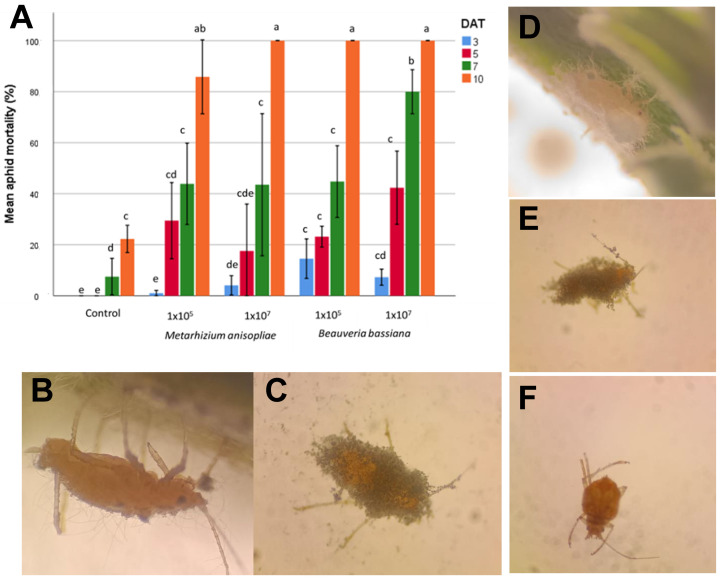
(**A**) Mean mortality (%) of *P. cannabis* recorded at different time intervals (DAT: Days after treatment) for conidial bioassays performed with *B. bassiana* and *M. anisopliae*. Treatments included two conidial concentrations (i.e., 1 × 10^5^ and 1 × 10^7^ conidia mL^−1^ and one control. Columns represent mean percent mortality ± SD (*n* = 10). The means followed by the same letters within columns are not significantly different from each other, according to Tukey’s HSD (*p* < 0.05). *Phorodon cannabis* infected with *Beauveria bassiana* 1 × 10^5^ (**B**), *Metarhizium anisopliae* 1 × 10^5^ (**C**), *Beauveria bassiana* 1 × 10^7^ (**D**), *Metarhizium anisopliae* 1 × 10^7^ (**E**), and uninfected control (**F**) on DAT 10 under a stereo microscope at 20× magnification.

**Figure 4 plants-14-00931-f004:**
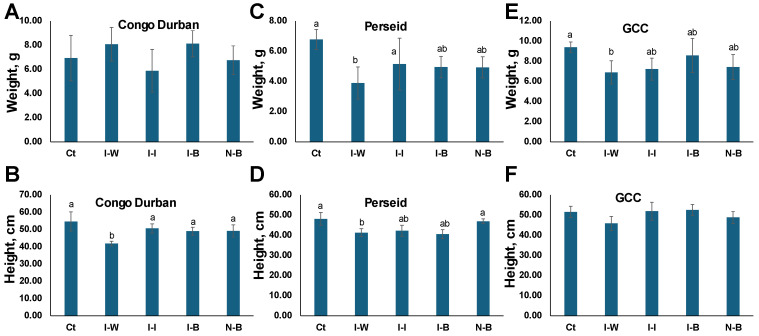
Growth parameters of Congo Durban (THC dominant), Perseid (CBD dominant), and GCC (balanced CBD/THC ratio) cannabis plants. (**A**,**C**,**E**) Dry biomass (g) after the flowering process and (**B**,**D**,**F**) Height (cm) of the plants (base of the plant to the highest point of the crown of the leaves). Ct—control; I-W—infested plants treated with water; I-I—infested plants treated with insecticide; I-B—infested plants treated with *B. bassiana* (1 × 10^7^ conidia); N-B—non-infested plants treated with *B. bassiana* (1 × 10^7^ conidia). Variables are expressed as the mean ± standard deviation. Means followed by same letter are not significantly different by Tukey’s HSD multiple range test at *p* < 0.05.

**Figure 5 plants-14-00931-f005:**
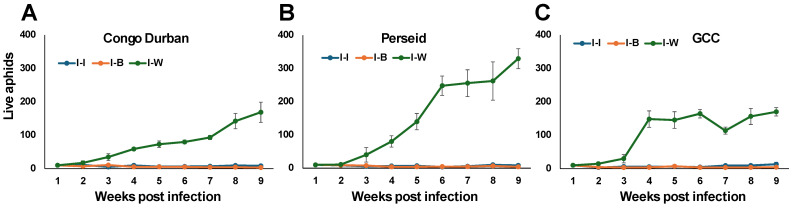
Effect of different treatments on the aphid *P. cannabis* population in the cannabis variety Congo Durban (**A**), Perseid (**B**), and GCC (**C**) under greenhouse conditions. Data represent mean aphid population counts per treatment (±SE) over the experimental period of 9 weeks after infestation. Infested plants treated with insecticide (I-I); infested plants treated with *B. bassiana* 1 × 10^7^ conidia mL^−1^ (I-B); infested plants treated with distilled water (I-W).

**Figure 6 plants-14-00931-f006:**
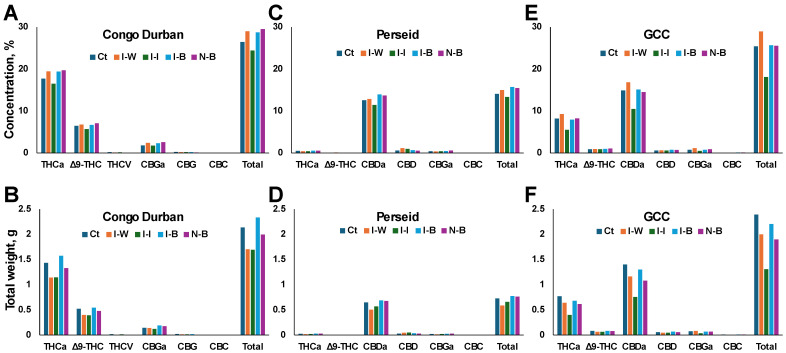
Cannabinoid profile of Congo Durban, Perseid, and GCC cannabis plants. Cannabinoid concentration (%): (**A**) Congo Durban, (**C**) Perseid, (**E**) GCC; total cannabinoids produced by a plant (g): (**B**) Congo Durban, (**D**) Perseid, (**F**) GCC. Cannabinoids were analyzed using USP <621> chromatography and HPLC-DAD quantification. Bb, *B. bassiana* 1 × 10^7^ conidia mL^−1^. Infested treated with insecticide (I-I), infested treated with Bb, *B. bassiana* 1 × 10^7^ conidia mL^−1^ (I-B), infested treated with distilled water (I-W), uninfested control (Ct), uninfested treated with *B. bassiana* 1 × 10^7^ conidia mL^−1^ (N-B).

**Figure 7 plants-14-00931-f007:**
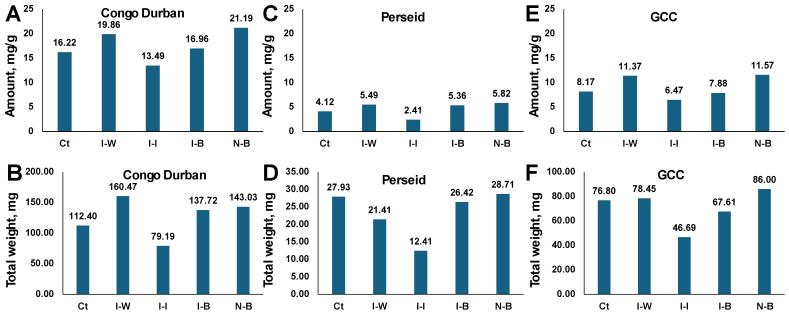
Total terpene concentration of different varieties of cannabis plants: potency and concentration analysis (%) using USP <621> chromatography and HPLC-DAD quantification. Terpene concentration in Congo Durban (**A**), Perseid (**C**), GCC (**E**). Total amount of terpenes produced per plant in Congo Durban (**B**), Perseid (**D**), GCC (**F**). Bb, *B. bassiana* 1 × 10^7^ conidia mL^−1^. Ct—control; I-W—infested plants treated with water; I-I—infested plants treated with insecticide; I-B—infested plants treated with *B. bassiana* (1 × 10^7^ conidia); N-B—non-infested plants treated with *B. bassiana* (1 × 10^7^ conidia).

## Data Availability

Data are available upon request.

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
