# Peer review of "Entomopathogenic Fungi Effectively Control Phorodon cannabis Aphid Population in Cannabis sativa Plants"

_plants, 2025, doi:10.3390/plants14060931_

Round 1
Reviewer 1 Report
Comments and Suggestions for Authors
Authors address the following comments.
1. What is the novelty of this research 2. Was the bioefficacy of strains/isolates of EPF tested in comparison with other established isolates? 3. Which method is followed for fungal culture purification? 4. Is ITS enough for EPF identification 5. Kindly provide the Accession number of Apids used in the experimentation. 6. In figure 7 & Figure 8..Kindly provide Standard error bar in bar diagramAuthor Response
- What is the novelty of this research
Our response: This research presents a novel approach to managing cannabis aphids (Phorodon cannabis) through the application of entomopathogenic fungi (EPF), specifically Beauveria bassiana. Given the limited existing studies on biological control of P. cannabis, this work significantly expands the knowledge base by demonstrating the efficacy of B. bassiana in reducing aphid populations while maintaining plant health. Furthermore, it uniquely explores the impact of aphid infestation and its biological control on the production of cannabinoids and terpenes in Cannabis sativa. The findings reveal that aphid infection alters secondary metabolite profiles, and while chemical insecticides suppress cannabinoid and terpene synthesis, B. bassiana treatment sustains or even enhances these compounds. This dual benefit—effective pest control and improved phytochemical quality—positions B. bassiana as a promising eco-friendly alternative to conventional insecticides in cannabis cultivation.
- Was the bioefficacy of strains/isolates of EPF tested in comparison with other established isolates?
Our response: No, we did not test the bioefficacy of strains/isolates of EPF tested in comparison with other established isolates. It would be interesting to do it in future research. We added this to the future directions.
- Which method is followed for fungal culture purification?
Our response: The single spore culture method was used to purify fungal cultures. A fungal suspension was prepared, and a sterile needle was dipped into the suspension and streaked onto a solid agar plate. The plates were incubated for several days to allow individual spores to develop into distinct colonies. Selected spores were then recultivated on a new solid plate to obtain purified fungal isolates. This information is now added to the Methods.
- Is ITS enough for EPF identification
Our response: The Internal Transcribed Spacer (ITS) region has recently been designated as the primary DNA barcode for the fungal kingdom. A DNA barcode is a short, highly variable, and standardized DNA sequence, approximately 700 nucleotides in length, used as a unique identifier for living organisms. The ITS region of nuclear rDNA is the most frequently sequenced marker for fungal taxonomy, enabling species-level identification and, in some cases, differentiation within species.
- Kindly provide the Accession number of Apids used in the experimentation.
Our response: The aphids used in the experimentation (Phorodon cannabis) were obtained from infected plants in Hepler Hall at the University of Lethbridge during the spring of 2023. However, no accession number is available, as their identification was based on morphological characterization using a stereo microscope at 20X magnification rather than molecular sequencing and deposition in a genetic database.
- In figure 7 & Figure 8..Kindly provide Standard error bar in bar diagram
Our response: Unfortunately, we are not able to provide error bars since we did not have enough resources to pay for the analysis of cannabinoids terpenes in multiple samples.
Reviewer 2 Report
Comments and Suggestions for Authors
Review of the manuscript „Newly isolated entomopathogenic fungi effectively control Phorodon cannabis aphid population in Cannabis sativa plants“
In my opinion, the manuscript brings interesting results especially concerning influence of fungi on the content of some chemicals in cannabis plants.
I have mainly formal comments to the manuscript.
- 33: why C in cannabis?
- 41: pathogens are transmitted, not the diseases (even when this formulation is frequently used).
- 54 and 63: the authors use two different names of the same organism. By the way, now its name is Akanthomyces lecanii (letter k is correct).
- 80 and many, many others: the authors almost consistently confuse the terms infection and infestation and infected/infested. The term infection (infected) belongs to pathogens, whereas the term infestation (infested) should be used for pests (and thus for aphids as well). Only exceptionally correct terms are used (eg. l. 337, 457, and 628).
- 186: treatments.
- 359: redundant comma after both species.
- 374: at a concentration of conidia.
- 384: missing full stop.
- 514: …for two… what? Minutes, hours…?
- 545-547: conidia and mycelia are plurals, so the verb were should be used.
- 588: inoculation is the term that should be used for pathogens. Aphids were transferred…
- 319 and 321: the word restored should be replaced by another word. There is no restoration in these cases, because each treatment was done separately, not in a series where some restoration of the content could take place.
The manuscript title is confusing for me. I would expect some really new species, not so notoriously known as B. bassiana and M. anisopliae.
For me the test of pathogenicity is not quite clear. If adult aphids were transferred to the treated leaf discs, they probably gave birth to some young aphids, especially in the case of control. So, how was the mortality calculated if the real number of aphids was practically unknown? Does it also mean that these young aphids were infected immediately after their birth? This should be mentioned at least in discussion.
The name of the insecticide used is not given. Thus, the information about its effect and comparison with other treatments is practically useless. At least its active ingredient should be mentioned.
When so common organisms as B. bassiana and M. anisopliae were isolated from dead aphids P. cannabis, why some commercial preparations of these fungi were not included in experiments to compare their effectivity with the isolates taken directly from P. cannabis?
Comments on the Quality of English Languageno
Author Response
- 33: why C in cannabis? - corrected
- 41: pathogens are transmitted, not the diseases (even when this formulation is frequently used). - corrected
- 54 and 63: the authors use two different names of the same organism. By the way, now its name is Akanthomyces lecanii (letter k is correct). – thank you, we have corrected it to the proposed name.
- 80 and many, many others: the authors almost consistently confuse the terms infection and infestation and infected/infested. The term infection (infected) belongs to pathogens, whereas the term infestation (infested) should be used for pests (and thus for aphids as well). Only exceptionally correct terms are used (eg. l. 337, 457, and 628). – you are absolutely right, we corrected it everywhere
- 186: treatments. - corrected
- 359: redundant comma after both species. – fixed/deleted
- 374: at a concentration of conidia. - corrected
- 384: missing full stop. – not sure about this one
- 514: …for two… what? Minutes, hours…? - corrected
- 545-547: conidia and mycelia are plurals, so the verb were should be used. - corrected
- 588: inoculation is the term that should be used for pathogens. Aphids were transferred… - corrected
- 319 and 321: the word restored should be replaced by another word. There is no restoration in these cases, because each treatment was done separately, not in a series where some restoration of the content could take place. – corrected
The manuscript title is confusing for me. I would expect some really new species, not so notoriously known as B. bassiana and M. anisopliae. – we agree and changed the title
For me the test of pathogenicity is not quite clear. If adult aphids were transferred to the treated leaf discs, they probably gave birth to some young aphids, especially in the case of control. So, how was the mortality calculated if the real number of aphids was practically unknown? Does it also mean that these young aphids were infected immediately after their birth? This should be mentioned at least in discussion.
Our response: We determined the number of live and dead aphids on each DAT to determine the total population and thus be able to calculate the percentage. Likewise, for the controls, the total population was determined to calculate the population reduction. If in any case there was asexual reproduction, given the DAT10 results of 100% infection, the fungal population was still present on the leaf and thus the conidia could come into contact with the juvenile aphids.
This is what we wrote in the methods: The petri dishes were incubated at room temperature (25°C) in the dark for 10 days. Each treatment was repeated ten times and data was taken on the mortality of the aphids on days 3, 5, 7 and 10 of incubation, during which time the dead aphids were removed, the number of dead and alive aphids was determined, and the dead aphids were then placed on a new PDA plate, incubated and then analyzed in the microscope to observe if the growth of the mycelium of the fungus was the cause of death.
The name of the insecticide used is not given. Thus, the information about its effect and comparison with other treatments is practically useless. At least its active ingredient should be mentioned.
Our response: This information is now provided.
When so common organisms as B. bassiana and M. anisopliae were isolated from dead aphids P. cannabis, why some commercial preparations of these fungi were not included in experiments to compare their effectivity with the isolates taken directly from P. cannabis?
Our response: This is a good idea, and unfortunately, we have not thought about it when planned experiments for the MSc student.
Reviewer 3 Report
Comments and Suggestions for Authors
This study reported two fungal isolates, Beauveria bassiana and Metarhizium anisopliae, and their bioactivity against Phorodon cannabis. I have the following suggestions.
!. The abstract should be re-written, clearly stated the main research data and results.
- Delete Table 1, replace Fig.1 with enough evidence for Beauveria and Metarhizium, besides colony.
- 20 fungal isolates were recovered,, colonies consistent with Trichoderma (F17, F18), Penicil- 9lium (F1), and Aspergillus (F10, F16) were identified, and only Two isolates, F2 and F12,were tested? Why?
- Figure 4.Add photos of Phorodon cannabis infectedwith different conidial concentrations and control. Provide more information for x- such as 1*105.
5.Line 185,treated with insecticide (I-I), which insecticide, more details.
- Fig. 5 and Fig. 7 provide more details in caption.
- Data analysis For Fig. 7 and 8?
- Table 2-4 as supplementary
- shorten the discussion and conclusion. Focused on your main results.
- delete data collection.
- Check the writing, such as Latin name, Line 41, entomopathogenic fungiand EPF
Author Response
- The abstract should be re-written, clearly stated the main research data and results.
Our response: We added this information to the Abstract.
- Delete Table 1, replace Fig.1 with enough evidence for Beauveria and Metarhizium, besides colony.
Our response: We have deleted Table 1. We moved Figure 1 into the supplemental material and changed the description in the Results section, indicating that morphological characteristics were supported by molecular analysis.
- 20 fungal isolates were recovered,, colonies consistent with Trichoderma (F17, F18), Penicil- 9lium (F1), and Aspergillus (F10, F16) were identified, and only Two isolates, F2 and F12,were tested? Why?
Our response: Among the 20 fungal isolates recovered, only F2 and F12 were tested because they were morphologically and molecularly identified as entomopathogenic fungi (EPF) from the genera Beauveria and Metarhizium, which are well-known for their insect-pathogenic properties. In contrast, the other isolates, such as Trichoderma, Penicillium, and Aspergillus, are not primarily entomopathogenic and were likely excluded from bioassays due to their lack of known insecticidal activity. The selection of F2 (Beauveria bassiana) and F12 (Metarhizium anisopliae) was based on their established effectiveness in biological control, particularly against aphid pests. Additionally, infection assays were conducted using the isolated fungi and aphids with the same leaf-dip method (data not shown), and only these two isolates (F2 - Beauveria bassiana and F12 - Metarhizium anisopliae) demonstrated activity against Phorodon cannabis.
- Figure 4. Add photos of Phorodon cannabis infected with different conidial concentrations and control. Provide more information for x- such as 1*105.
Our response: We added the pictures.
5.Line 185, treated with insecticide (I-I), which insecticide, more details.
Our response: Doktor Doom Premium Indoor/Outdoor 3-in-1 Crop & Plant Rescue Concentrate. This information is added now.
- Fig. 5 and Fig. 7 provide more details in caption.
Our response: Information added.
- Data analysis For Fig. 7 and 8?
Our response: We could not analyze these data statistically, as a single sample per each group was analyzed. We did not have enough resources to pay ~$200 per sample.
- Table 2-4 as supplementary
Our response: This was moved to Supplementary material.
- shorten the discussion and conclusion. Focused on your main results.
Our response: We substantially shortened both sections.
- delete data collection.
Our response: Deleted.
- Check the writing, such as Latin name, Line 41, entomopathogenic fungi and EPF
Our response: We rechecked and corrected the grammar.
Reviewer 4 Report
Comments and Suggestions for Authors
The manuscript adds to the body of literature on the use of entomopathogenic fungi in pest control. Although the science has merit, I struggled with the flow of the manuscript.
- Is it the convention of the journal to have the Methods section just before the conclusions? This is odd.
- The authors should have performed Koch's postulate to confirm the causal agent of the mortality of the aphids.
- In terms of logical flow, I would have expected a section on the impact of the aphids on plant growth in the introduction as a motivation for the study. Then the flow should be: isolation and characterization of the fungi, dose response bioassay followed by a time response bioassay. This will then allow a conclusion of which fungus to use and at what concentration. All of the information on the impact of the aphids on terpenes for the different varieties and how aphid infect affects growth in different varieties is probably sufficient for another paper. The aim of this paper is the possible role of EPF for aphid control in cannabis.

Author Response
- Is it the convention of the journal to have the Methods section just before the conclusions? This is odd.
Our response: Yes, this is the template provided by the journal.
- The authors should have performed Koch's postulate to confirm the causal agent of the mortality of the aphids.
Our response: Each treatment was repeated ten times and data was taken on the mortality of the aphids on days 3, 5, 7 and 10 of incubation, during which time the dead aphids were removed, placed on a new PDA plate, incubated and then analyzed in the microscope to observe if the growth of the mycelium of the fungus was the cause of death.
We understand that it would be ideal to isolate the fungus from dead aphid, and to reinfect other aphids, but this was how we isolated the strains in the beginning.
- In terms of logical flow, I would have expected a section on the impact of the aphids on plant growth in the introduction as a motivation for the study. Then the flow should be: isolation and characterization of the fungi, dose response bioassay followed by a time response bioassay. This will then allow a conclusion of which fungus to use and at what concentration. All of the information on the impact of the aphids on terpenes for the different varieties and how aphid infect affects growth in different varieties is probably sufficient for another paper. The aim of this paper is the possible role of EPF for aphid control in cannabis.
Our response: We added section of the impact of aphids. We think that splitting the paper in two would make both of them incomplete, unless we do a lot more experimental work.
We also took care of points in the pdf, including adjusting the Y axis to be equal for all cultivars.
Round 2
Reviewer 1 Report
Comments and Suggestions for Authors
Accept the current form of MS
Author Response
Thanks
Reviewer 2 Report
Comments and Suggestions for Authors
Review of the revised manuscript Entomopathogenic fungi effectively control Phorodon cannabis aphid population in Cannabis sativa plants
Mostly, the errors have been corrected (by items).
Unfortunately, this does not apply to the problem of the terms infection/infestation and infected/infested. It still persists eg. on lines 210, 242, 245,272, 296, 317, 322, 334, 441, 487, 531, fig. 5 (description of graph axes) and possibly on others. On the other hand, on line 194 (fig. 4F) correct uninfected was erroneously changed to uninfested. This picture is the aphid that is not infected by the fungus. The authors have to realize that especially in the text combining infection of aphids by fungi and infestation of plants by aphids the correct terms are essential to prevent confusion.
- 52: in my opinion, ants cannot be generally taken as plant pests, because they do not directly harm plants.
- 99: the term “strain” should be used for some population that is at least partly characterized. Here the term isolate (eg. i. was obtained) should be used. On the other hand, on l. 77 the term strain is used correctly, because in commercial products characterized populations are used.
- 177: at the highest conidia concentration, or 1x107 conidia/mL-1).
Fig. 3: I doubt meaning of the addition of other photos of aphids since they are not of a good quality and no details are clearly seen. The photos seem to be taken by transmission light microscope and not by a binocular with upper light where it should be better visible.
- 570: Akanthomyces.
- 652: each of the plants.
Concerning the insecticide used, commercial name was added. However, I was unable to find this concrete insecticide on the website of Doktor Doom. So, active ingredient (perhaps pyrethrine) should be given as well.
Author Response
Reviewer 2: Mostly, the errors have been corrected (by items).
Unfortunately, this does not apply to the problem of the terms infection/infestation and infected/infested. It still persists eg. on lines 210, 242, 245,272, 296, 317, 322, 334, 441, 487, 531, fig. 5 (description of graph axes) and possibly on others. On the other hand, on line 194 (fig. 4F) correct uninfected was erroneously changed to uninfested. This picture is the aphid that is not infected by the fungus. The authors have to realize that especially in the text combining infection of aphids by fungi and infestation of plants by aphids the correct terms are essential to prevent confusion.
Our response:
Thank you, we fully agree, and hopefully, this time, we addressed them all.
- 52: in my opinion, ants cannot be generally taken as plant pests, because they do not directly harm plants.
Our response: We changed it to “insects”, but we actually disagree in this case. Specifically, when ants are attacked to honeydew released by aphids, they are considered pests.
- 99: the term “strain” should be used for some population that is at least partly characterized. Here the term isolate (eg. i. was obtained) should be used. On the other hand, on l. 77 the term strain is used correctly, because in commercial products characterized populations are used.
Our response: Hmm, in the beginning of the Results section, we describe the process of isolation of various fungi prior to their characterization, so we used the term “isolate”. We corrected “isolate” to “strain” in line 77.
- 177: at the highest conidia concentration, or 1x107 conidia/mL-1).
Fig. 3: I doubt meaning of the addition of other photos of aphids since they are not of a good quality and no details are clearly seen. The photos seem to be taken by transmission light microscope and not by a binocular with upper light where it should be better visible.
Our response: this was added in response to the request by another reviewer.
- 570: Akanthomyces.
Our response: corrected
- 652: each of the plants.
Our response: corrected
Concerning the insecticide used, commercial name was added. However, I was unable to find this concrete insecticide on the website of Doktor Doom. So, active ingredient (perhaps pyrethrine) should be given as well.
Out response: The active ingredient is permethrin. This information is now updated.
Reviewer 3 Report
Comments and Suggestions for Authors
This revised version has improved significantly and I suggest to accept it.
Author Response
Thanks
Reviewer 4 Report
Comments and Suggestions for Authors
The authors have addressed all of my comments
Author Response
Thanks